# High Thermal Dissipation of Normally off p-GaN Gate AlGaN/GaN HEMTs on 6-Inch N-Doped Low-Resistivity SiC Substrate

**DOI:** 10.3390/mi12050509

**Published:** 2021-05-01

**Authors:** Yu-Chun Huang, Hsien-Chin Chiu, Hsuan-Ling Kao, Hsiang-Chun Wang, Chia-Hao Liu, Chong-Rong Huang, Si-Wen Chen

**Affiliations:** Department of Electronics Engineering, Chang Gung University, Taoyuan 33324, Taiwan; james19961202@gmail.com (Y.-C.H.); snoopy@mail.cgu.edu.tw (H.-L.K.); smallflgt@hotmail.com (H.-C.W.); r3287133@gmail.com (C.-H.L.); gain525252@gmail.com (C.-R.H.); swchen@mail.cgu.edu.tw (S.-W.C.)

**Keywords:** p-GaN gate HEMT, normally off, low-resistance SiC substrate, temperature

## Abstract

Efficient heat removal through the substrate is required in high-power operation of AlGaN/GaN high-electron-mobility transistors (HEMTs). Thus, a SiC substrate was used due to its popularity. This article reports the electrical characteristics of normally off p-GaN gate AlGaN/GaN high-electron-mobility transistors (HEMTs) on a low-resistivity SiC substrate compared with the traditional Si substrate. The p-GaN HEMTs on the SiC substrate possess several advantages, including electrical characteristics and good qualities of epitaxial crystals, especially on temperature performance. Additionally, the price of the low-resistivity SiC substrate is three times lower than the ordinary SiC substrate.

## 1. Introduction

Gallium nitride high-electron-mobility transistors (HEMTs) have attracted increasing attention in the field of high-frequency and high-power device applications due to their high breakdown field, high mobility, and good thermal properties. However, they are naturally normally on devices. For high-power applications, off devices are normally desirable for system reliability [1]. Thus, several research works have been proposed to realize the normally off operation characteristics of AlGaN/GaN HEMTs such as fluorine base plasma treatment [2,3], an ultrathin barrier (UTB) [4], and gate-recessed structures [5,6].

Recently, GaN HEMTs with a p-GaN gate stack (p-GaN gate HEMTs) have been suggested as one of the candidates, in which a p-GaN layer on top of the AlGaN barrier depletes the 2D electron gas carriers in the channel [7,8,9]. The normally off p-GaN gate AlGaN/GaN high-electron-mobility transistor (HEMT) on a SiC substrate is expected to be a good choice for high-power switching components due to its high thermal conductivity, low resistivity, and high-voltage capability. Another advantage of using a SiC substrate is its lower lattice mismatch of ~3% for GaN (that of Si is ~17%). Owing to the high material properties of gallian nitride and the SiC substrate, these devices are expected to operate in high-temperature environments [10].

Here, we analyzed the DC, breakdown, pulsed, and thermal measurement performances of AlGaN/GaN HEMTs with a p-GaN gate between low-resistivity SiC and ordinary Si substrates. Finally, the heat removal through the SiC substrate had the most outstanding performance.

## 2. Experimental Procedures

In this work, an epitaxy wafer was grown by metal organic chemical vapor deposition on 6-in n-doped low-resistivity SiC substrates, as shown in Figure 1a. A 650-nm-thick undoped GaN channel layer was grown on top of a 3.8-µm-thick undoped GaN buffer layer. A 17.5-nm-thick undoped barrier (1.5-nm-AlN/15-nm-AlGaN/1-nm-AlN) layer was sandwiched between the GaN channel layer and a 75-nm p-type GaN cap layer. The fabrication process started with device isolation by an Ar implantation. The 5-μm-long p-GaN gate island was removed by Cl_2_/BCl_3_/SF_6_ dry etching and the etching depth was stopped by the 1 nm AlN etching stop layer. After dry etching, the source and drain ohmic contacts were prepared using the electron beam evaporation of a multilayered Ti/Al/Ni/Au (25 nm/120 nm/25 nm/150 nm) sequence, patterned by a lift-off process, and annealed by a 30-s rapid thermal annealing (RTA) at 875 °C in ambient N_2_. Finally, a Ti/Au (25/150 nm) gate metal stack was deposited, and 100 nm of SiN was passivated by plasma-enhanced chemical vapor deposition (PECVD). The descriptions above were made by our own laboratory research process.

## 3. Results and Discussion

Figure 2a,b show the log-scale transfer (I_DS_-V_GS_) and output (I_DS_-V_DS_) characteristics of GaN on a low-resistivity SiC substrate HEMT (LRSiC-HEMT) and a Si substrate HEMT (Si-HEMT). As shown in Figure 2a, the off-state currents for the LRSiC-HEMT and Si-HEMT are 1.37 × 10^−5^ and 5.2 × 10^−5^ mA/mm at V_GS_ = 0 V, the I_on_/I_off_ ratios are 1.5 × 10^8^ and 1.85 × 10^6^, and they deliver the normally off operation with a positive V_TH_ of 3.2 V and 1.8 V defined at I_D_ = 1 mA/mm, respectively. In Figure 2b, the corresponding maximum drain current density (I_Dmax_) values are 131 mA/mm and 108 mV/mm at a gate-to-source voltage (V_GS_) = 8 V and a drain-to-source voltage (V_DS_) = 10 V. The I_Dmax_ value of the LRSiC-HEMT was 28% higher than that of the Si-HEMT. Additionally, the LRSiC-HEMT also exhibits a lower on resistance (R_ON_) of 16 Ω·mm.

The breakdown voltage, BV, of the devices is determined by the drain leakage current reaching 1 mA/mm. As shown in Figure 3, the off-state breakdown voltages and vertical breakdown voltages of LRSiC-HEMT and Si-HEMT are 325 V, 310 V, 413 V, and 319 V, respectively. Vertical breakdown voltage measurements were performed on both wafers by the grounded substrate, and the ohmic contact pattern was swept from 0 V up to the breakdown voltage; the size of the ohmic pattern is about 100 × 100 μm. Although the epitaxy technology using the low-resistivity SiC substrate was not as stable, the on device performance and the substrate’s breakdown voltages were still better than the traditional Si substrate.

Having confirmed the importance of the surface temperature distribution measurements, the temperature–time curve is shown in Figure 4 by using an infrared (IR) thermographic system with micro-Raman spectroscopy. On account of having much higher thermal conductivity, SiC could achieve an outstanding performance on heat dissipation. For the set of device measurements, V_DS_ was held constant at 10 V, while I_DS_ was kept at 100 mA/mm for 60 s and then waited for the device cool down for about 50 s. As expected, the center of the gate area reached the highest temperature. As shown in Figure 4, following the time, the temperature gradient appears to increase. The temperature of LRSiC-HEMT was heated up to 34.4 °C and cooled down to 30 °C rapidly. On the contrary, Si-HEMT accumulated much more heat to reach 37.5 °C and removed it slowly.

To investigate the thermal stability of LRSiC-HEMT, transfer characteristics were measured from room temperature (25 °C) to 175 °C with a 50 °C step (Figure 5a). The device shows a good thermal stability with the V_TH_ shifting less than 0.4 V up to 175 °C (Figure 5b). R_ON_ increases by about 3.3 times, due to stronger phonon scattering at higher temperature [11].

To analyze the trapping/detrapping effect, the pulsed I–V characteristic and the dynamic R_on_ ratio of LRSiC-HEMT were measured using a pulse width of 2 μs and a period of 200 μs. The carriers will be trapped in the buffer layer or near the surface which is in the AlGaN layer or passivation interface during the pulse measurement. Therefore, it will reduce the carrier density and make the resistance higher [12]. In this work, the V_DSQ_ for LRSiC-HEMT is swept from 0 to 80 V with a step of 20 V in Figure 6. Clearly, the current collapse and the dynamic R_ON_ ratio of Si-HEMT are both worse than LRSiC-HEMT. Additionally, being under relatively high stress led to an I–V slope decrease because of the defect trap density in the buffer layer and surface. In Figure 7, the dynamic R_ON_ ratios are 6.8 times and 2.5 times at V_DSQ_ = 80 V.

## 4. Conclusions

In this work, normally off AlGaN/GaN high-electron-mobility transistors (HEMTs) with a p-GaN gate on a low-resistivity SiC substrate (LRSiC-HEMT) were developed. Comparing to Si-HEMT, LRSiC-HEMT obtained many advantages, such as a higher output current, higher off-state and vertical breakdown voltages, and a lower dynamic Ron ratio, especially in thermal performance. In addition, the price of the low-resistivity SiC substrate is much lower than the ordinary SiC substrate, which is shown in Table 1. Therefore, it holds promise to be an excellent choice to solve the heat problem and cost consideration for power devices.

## Figures and Tables

**Figure 1 micromachines-12-00509-f001:**
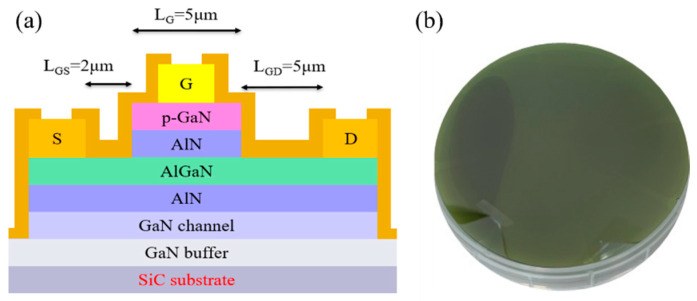
(**a**) Structure of p-GaN gate HEMT on low-resistance substrate. (**b**) Outward appearance of low-resistance SiC wafer.

**Figure 2 micromachines-12-00509-f002:**
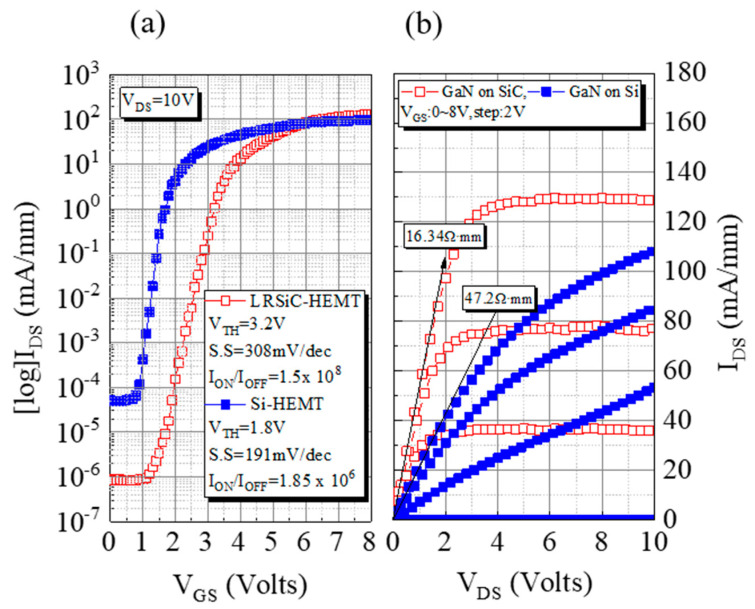
I–V characteristics of LRSiC-HEMT and Si-HEMT with L_GS_/L_G_/L_GD_/W_G_ = 2/5/5/100 μm. (**a**) Transfer I_DS_-V_GS_ characteristic. (**b**) Output I_DS_-V_DS_ characteristic.

**Figure 3 micromachines-12-00509-f003:**
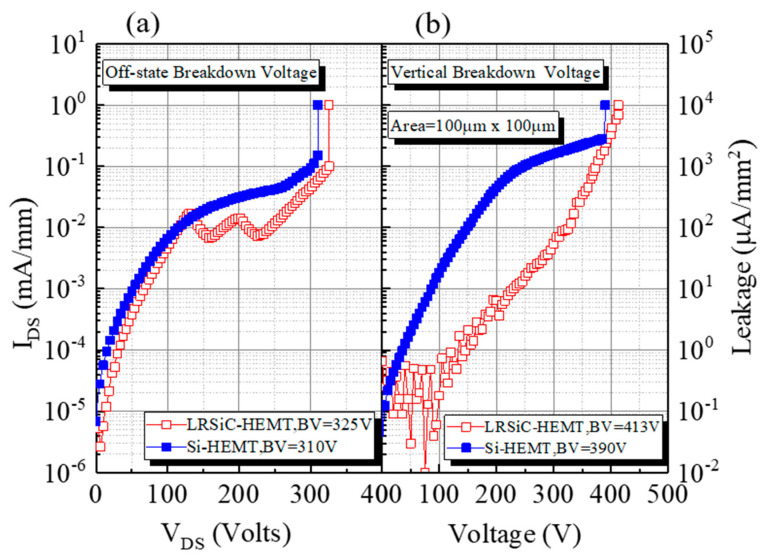
(**a**) Off-state breakdown voltage and (**b**) vertical breakdown voltage measurement.

**Figure 4 micromachines-12-00509-f004:**
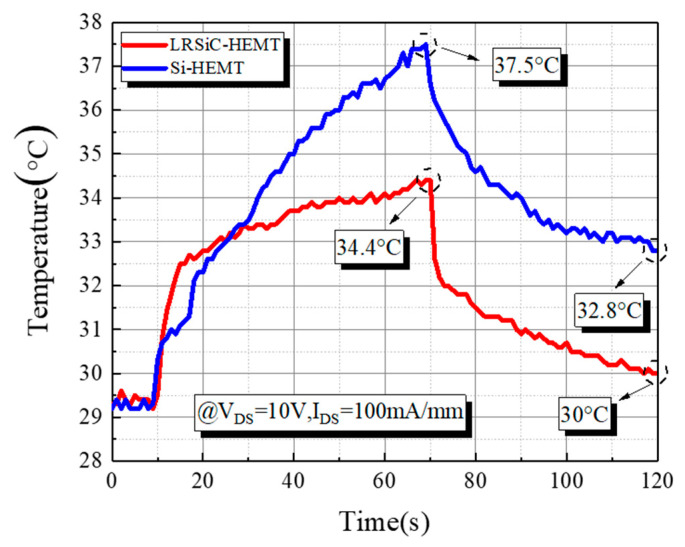
Temperature following the time, with the device operating for 60 s and cooling down for 50 s.

**Figure 5 micromachines-12-00509-f005:**
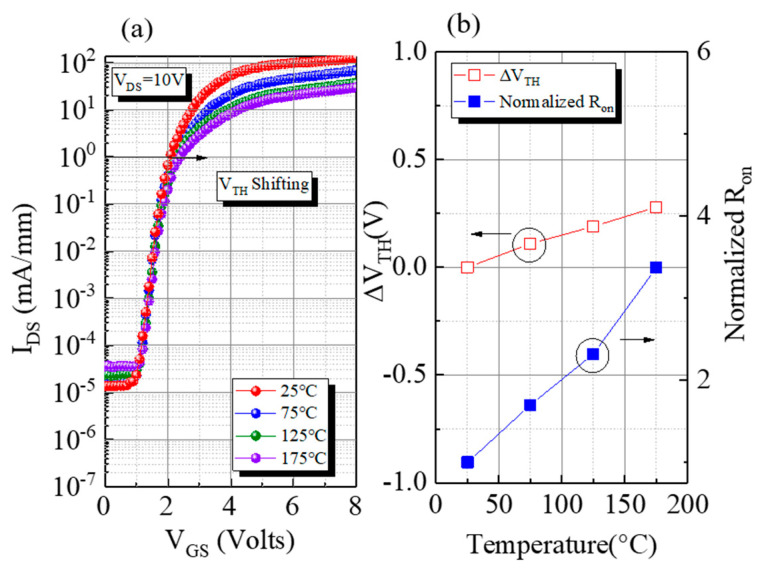
(**a**) Transfer characteristics of LRSiC-HEMT from 25 to 175 °C; (**b**) T-dependence of V_TH_ and R_ON_.

**Figure 6 micromachines-12-00509-f006:**
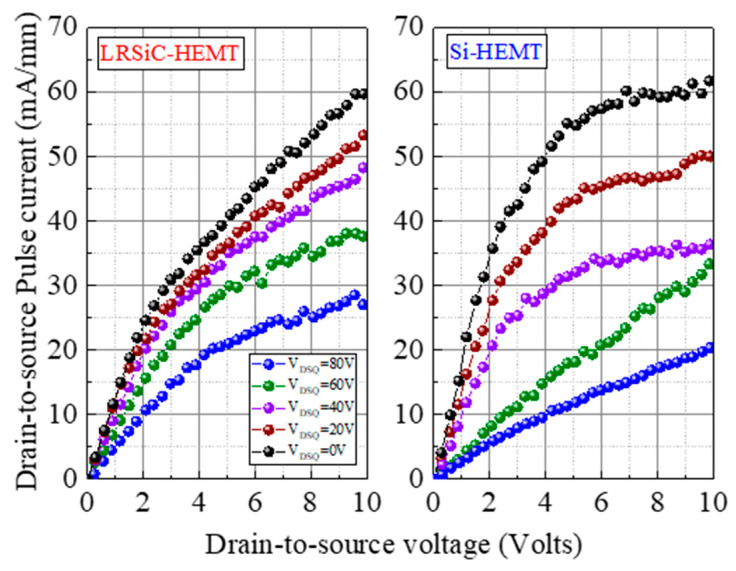
Pulsed I_DS_−V_DS_ characteristics from quiescent gate bias (V_GSQ_) point of 0 V with 2 µs pulse width and 200 µs pulse period. The quiescent drain bias (V_DSQ_) was then swept from 0 to 80 V (in 20-V increments).

**Figure 7 micromachines-12-00509-f007:**
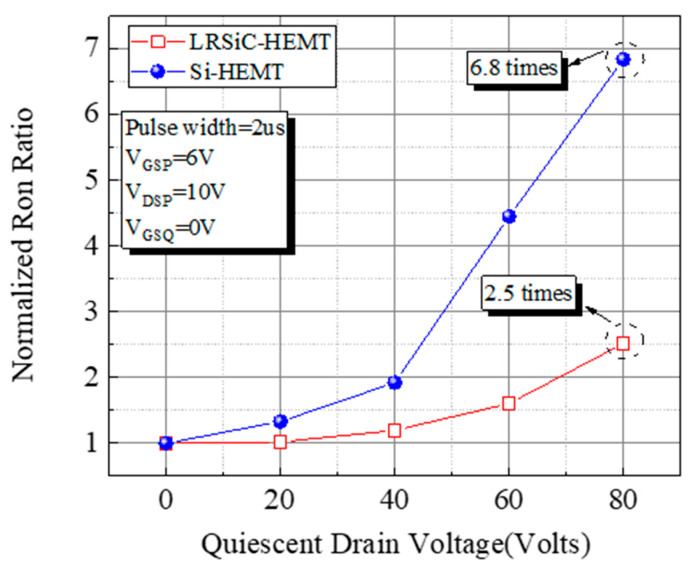
Dynamic R_on_ ratio of LRSiC-HEMT and Si-HEMT.

**Table 1 micromachines-12-00509-t001:** Reference price and resistivity of low-resistivity SiC substrate and high-resistivity SiC substrate.

	Reference Price	Resistivity (Ω·cm)
LRSiC (6 inch)	USD1000	0.015~0.025
HRSiC (6 inch)	USD3000	>1E5

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
