# Peer review of "High Thermal Dissipation of Normally off p-GaN Gate AlGaN/GaN HEMTs on 6-Inch N-Doped Low-Resistivity SiC Substrate"

_micromachines, 2021, doi:10.3390/mi12050509_

Round 1
Reviewer 1 Report
Thank you to the authors for a well written manuscript.
My comments are as follows:
- The major topic of the paper is the development of a GaN-on-SiC pGaN HEMT using a low resistivity SiC substrate. As such I believe more emphasis should be given in the introduction on what is new about the work in this study. Have GaN-on low resistivity SiC power devices been reported in literature previously? If so, how does the performance of the devices in this study compare with previously reported devices.
- It is not very clear to me whether the authors performed the GaN-on-SiC and GaN-on-Si epitaxy or whether they purchased the wafers after the epitaxy and only performed the device fabrication process. Please clarify this in the experimental procedure.
- What was the vertical breakdown of the wafers used intended to be?
- Are there any challenges of extending the off-state rating of the LRSIC-HEMT to 650V (which is a rich market for GaN-on-Si HEMTs) compared to an ordinary GaN-on-SiC HEMT or a GaN-on-Si HEMT?
- Beyond breakdown are there any other challenges for LRSiC-HEMT compared to ordinary GaN-on-SiC HEMT? I think it would be good to outline why using a low resistivity SiC substrate is not the norm despite the clear cost advantage over an ordinary SiC substrate.
- The GaN-on-Si HEMT used for comparison is not a very competitive device in terms of what is available, even in qualified processes by high volume foundries [see reference below from 2015 for example]. Therefore, I believe the LRSiC HEMT appears good in comparison to a GaN-on-Si HEMT which is not state of the art.
- How to you explain the significantly higher threshold voltage in the LRSiC-HEMT?
K-Y. R. Wong et al., "A next generation CMOS-compatible GaN-on-Si transistors for high efficiency energy systems," 2015 IEEE International Electron Devices Meeting (IEDM), Washington, DC, USA, 2015, pp. 9.5.1-9.5.4, doi: 10.1109/IEDM.2015.7409663.
Author Response
Reviewer1
- The major topic of the paper is the development of a GaN-on-SiC pGaN HEMT using a low resistivity SiC substrate. As such I believe more emphasis should be given in the introduction on what is new about the work in this study. Have GaN-on low resistivity SiC power devices been reported in literature previously? If so, how does the performance of the devices in this study compare with previously reported devices.
Currently, there’s no study about p-GaN gate AlGaN/GaN HEMTs on the low resistivity SiC substrate. In general, the crystal angle of SiC substrate were usually 4 degrees. However, the low resistivity SiC substrates we used in this study were 0 degree. As we know, it’s the first time demonstration of p-GaN Gate AlGaN/GaN HEMTs on the new type substrate. Fig.1. shows the outward appearance of the devices on 6-inch ow Resistivity SiC Substrate. We believe this demonstration and related improved characteristics were beneficial for the next generation GaN E-mode device possible development trend.
- It is not very clear to me whether the authors performed the GaN-on-SiC and GaN-on-Si epitaxy or whether they purchased the wafers after the epitaxy and only performed the device fabrication process. Please clarify this in the experimental procedure.
Thank you for your kind reminder. Actually, we purchased the substrate then epitaxy and fabrication were made by our own research process.
- What was the vertical breakdown of the wafers used intended to be?
We expected the vertical leakage current could lower than 1μA/mm2.
- Are there any challenges of extending the off-state rating of the LRSIC-HEMT to 650V (which is a rich market for GaN-on-Si HEMTs) compared to an ordinary GaN-on-SiC HEMT or a GaN-on-Si HEMT?
Power devices operate at different breakdown voltages, which can be roughly divided into high-voltage 650V, medium-voltage 100V, and low-voltage 40V. Since it’s the first time using on the new type substrate, there are still some points need to overcome. Including epitaxy quality and the process recipe. However, LRSiC HEMT has already complied with a standard in medium-voltage and low-voltage.
- Beyond breakdown are there any other challenges for LRSiC-HEMT compared to ordinary GaN-on-SiC HEMT? I think it would be good to outline why using a low resistivity SiC substrate is not the norm despite the clear cost advantage over an ordinary SiC substrate.
Nitrogen was used as doping source during the LRSiC substrate production process. According to the X-ray diffraction analysis, the peak of N-doped SiC shift to a higher wavenumber which means N-doped SiC implies that the amount of vacancy defects formed in N-doped SiC is larger than that in undoped SiC. Apparently, the N doping increases the amount of defects in SiC[1].
[1] Y.K. Dou, J.B. Li, X.Y. Fang, H.B. J, and M.S. Cao, “The enhanced polarization relaxation and excellent high-temperature dielectric properties of N-doped SiC,” Appl. Phys. Lett. 104, 052102 (2014)
- The GaN-on-Si HEMT used for comparison is not a very competitive device in terms of what is available, even in qualified processes by high volume foundries [see reference below from 2015 for example]. Therefore, I believe the LRSiC HEMT appears good in comparison to a GaN-on-Si HEMT which is not state of the art.
Thank you for your valuable insights and agreements. We also expect the development of LRSiC HEMT be widely used in the future.
- How to you explain the significantly higher threshold voltage in the LRSiC-HEMT?
Since SiC has good heat dissipation, we found that the activation of Mg in pGaN were relatively enhanced during 750 oC activation environment. Higher Mg concentration leads to a more positive. According to the C-V measurement, GaN on SiC really achieved the lager capacitance.
Reviewer 2 Report
Thank you to the authors for this submission. I have a few areas of improvement.
- You should describe where your devices were fabricated. Is it an industrial fab, or your own research process? The paper just states that the devices were fabricated and gives some steps, whereas more detail about where and how they were made would be important to include.
- The comparison of what the experiment here is isn’t clear. It seems to be an iterative improvement on an existing device, but that isn’t clearly stated.
- There is no description of the measurement technique used to collect thermal data. Specifically Figure 4 shows this time transient thermal data without any explanation as to how it was measured. This looks somewhat similar to a device I have seen: T3STER . https://www.plm.automation.siemens.com/global/en/products/simcenter/t3ster.html Is this what you used?
- This reference may be of use to you which measured temperature of GaN HEMT channels by another method: https://ieeexplore.ieee.org/stamp/stamp.jsp?arnumber=7517727
- I like the paper, but overall it could use more detail to increase its value. This should be done by more thoroughly explaining the experiment or comparison being made, the fabrication changes included in that experiment, and describing measurement techniques. Without this information, the paper will not be useful to many people, and negates the need to publish it.
Author Response
Reviewer2
- You should describe where your devices were fabricated. Is it an industrial fab, or your own research process? The paper just states that the devices were fabricated and gives some steps, whereas more detail about where and how they were made would be important to include.
The devices were fabricated by our own research process. Additionally, the epitaxy of structure were added in the experimental procedure. And the whole fabrication process can be divided into five steps in descending order are implantation, p-GaN gate island etching, ohmic contact, p-GaN metal stack and SiN passivate.
- The comparison of what the experiment here is isn’t clear. It seems to be an iterative improvement on an existing device, but that isn’t clearly stated.
The main ideal of the research is using the SiC substrate to replace Si substrate because SiC has less lattice mismatch and better heat dissipation. However, we use the low resistivity SiC substrate base on cost considerations.
- There is no description of the measurement technique used to collect thermal data. Specifically Figure 4 shows this time transient thermal data without any explanation as to how it was measured. This looks somewhat similar to a device I have seen: T3STER . https://www.plm.automation.siemens.com/global/en/products/simcenter/t3ster.htmlIs this what you used?
Thank you for your kind reminder. I have corrected the description about the Figure. 4. It should be surface temperature distribution measurements by infrared(IR) thermographic system with micro-Raman spectroscopy.
- This reference may be of use to you which measured temperature of GaN HEMT channels by another method: https://ieeexplore.ieee.org/stamp/stamp.jsp?arnumber=7517727
Thank you for your nice suggestion, it will be very helpful for us. We will spend time to comprehend the contents.
- I like the paper, but overall it could use more detail to increase its value. This should be done by more thoroughly explaining the experiment or comparison being made, the fabrication changes included in that experiment, and describing measurement techniques. Without this information, the paper will not be useful to many people, and negates the need to publish it.
Thank you for your detailed explanation, the suggestions above are added to the paper. We also expect the paper to be published. Believing that after the technique become mature, it will be an excellent choice for power devices.
